# A limbic circuit selectively links active escape to food suppression

Estefania P Azevedo[1], Bowen Tan[1], Lisa E Pomeranz[1], Violet Ivan[1], Robert Fetcho[2], Marc Schneeberger[1], Katherine R Doerig[1], Conor Liston[2], Jeffrey M Friedman[1,3]*, Sarah A Stern[1]*

[1]Laboratory of Molecular Genetics, The Rockefeller University, New York, United States; [2]Department of Psychiatry, Weill Cornell Medical College-New York Presbyterian Hospital, New York, United States; [3]Howard Hughes Medical Institute, The Rockefeller University, New York, United States

**Abstract** Stress has pleiotropic physiologic effects, but the neural circuits linking stress to these responses are not well understood. Here, we describe a novel population of lateral septum neurons expressing neurotensin (LS$^{Nts}$) in mice that are selectively tuned to specific types of stress. LS$^{Nts}$ neurons increase their activity during active escape, responding to stress when flight is a viable option, but not when associated with freezing or immobility. Chemogenetic activation of LS$^{Nts}$ neurons decreases food intake and body weight, without altering locomotion and anxiety. LS$^{Nts}$ neurons co-express several molecules including Glp1r (glucagon-like peptide one receptor) and manipulations of Glp1r signaling in the LS recapitulates the behavioral effects of LS$^{Nts}$ activation. Activation of LS$^{Nts}$ terminals in the lateral hypothalamus (LH) also decreases food intake. These results show that LS$^{Nts}$ neurons are selectively tuned to active escape stress and can reduce food consumption via effects on hypothalamic pathways.

*For correspondence:
friedj@mail.rockefeller.edu (JMF);
sstern01@rockefeller.edu (SAS)

**Competing interests:** The authors declare that no competing interests exist.

## Introduction

A key function of the central nervous system (CNS) is to sense external conditions and orchestrate an appropriate behavioral response (*McEwen and Akil, 2020*). The detection of stressful conditions, in particular, is of critical importance as it enables animals to remove themselves from dangerous situations while suppressing competing (maladaptive) behaviors. Responses to stress can differ and can be characterized as either active coping (e.g. fight or flight) or passive coping (e.g. freezing, immobility). These responses are also associated with other adaptive behavioral effects including reduced feeding until the period of danger has passed, and these stereotyped feeding responses are conserved across organisms ranging from flies to mice and humans (*Yau and Potenza, 2013*; *Surendran et al., 2017*). Although an association between stress and reduced food intake is well established, the specific neural circuits that are activated by active versus passive coping strategies and reduced food intake have not been delineated.

We, therefore, set out to map neural circuits that are activated by stress and regulate feeding by identifying the neuronal cell types that are activated after immobilization (i.e. acute restraint) using a well-established behavioral paradigm that leads to a robust decrease in food intake for up to 48 hr (*Donohoe, 1984*; *Shimizu et al., 1989*). Previous studies have shown increased expression of c-fos, an early immediate gene, in many areas of the brain following acute stress, including regions of the limbic system (e.g. lateral septum; *Sheehan et al., 2004*; *Aloisi et al., 1997*; *Figueiredo et al., 2002*; *Sood et al., 2018*). Despite this finding, neither the cell-type specificity and the activation dynamics of stress-responsive neurons in the limbic system have been determined nor the mechanisms by which these neurons regulate homeostatic behaviors, such as feeding.

In this study, we used an unbiased transcriptomic approach (*Knight et al., 2012*) to map and identify neurons in the limbic system that are activated during stressful situations. Our studies revealed that neurons in the lateral septum that express neurotensin (LS$^{Nts}$ neurons) are acutely activated by restraint stress. In vivo calcium recordings from LS$^{Nts}$ neurons show that these neurons selectively respond to stressful situations when escape is a viable option, indicating that these neurons are selectively tuned to active coping. Acute or chronic chemogenetic activation of these neurons leads to significant reductions in food intake and body weight, without an effect on locomotion and anxiety-like behaviors. Molecular profiling of LS$^{Nts}$ neurons revealed co-expression of glucagon-like-peptide one receptor (*Glp1r*) and accordingly, activation of LS$^{Glp1r}$ neurons also led to a decrease in feeding. Finally, we mapped the projections of the LS$^{Nts}$ neurons and found that optogenetic activation of LS$^{Nts}$ projections to the lateral hypothalamus also reduces feeding in mice. Overall, our data define a limbic circuit selectively tuned to active escape stress and that, once activated, can reduce food intake via projections to the lateral hypothalamus, a well-established feeding center.

## Results

### Identification of a limbic population activated by stress

To identify specific neuronal populations that respond to acute stress, we subjected wild-type C57Bl/6J mice to restraint stress using a flexible, plastic decapicone for immobilization. After 1 hr, we mapped the expression pattern of c-fos in the whole mouse brain and found increased numbers of c-fos+ cells in several regions of the brain such as the basolateral amygdala (BLA), bed nucleus of the stria terminalis (BNST), cortex (insular cortex [IC]), lateral septum (LS; *Figure 1A*), and elsewhere (*Supplementary file 1*). Consistent with previous studies (*Kubo et al., 2002*), acute restraint stress led to a marked increase in the number of c-fos+ cells in the LS compared to controls (*Figure 1—figure supplement 1A*, Unpaired Student's t-test, \*p<0.05). The LS is part of the anterior limbic system and is known to regulate several emotional behaviors, particularly stress, anxiety, and aggression (*Kubo et al., 2002*; *Anthony et al., 2014*; *Wong et al., 2016*). In addition, a previous study from our group and another have shown that optogenetic activation of neurons in the LS has a negative effect on feeding (*Azevedo et al., 2019*; *Sweeney and Yang, 2016*), raising the possibility that specific neurons in this region might link restraint stress to a non-homeostatic reduction in food intake.

Next, we set out to determine the molecular identity of the neurons in the LS that are activated by acute restraint stress using PhosphoTrap, an unbiased transcriptomic method to molecularly profile activated neurons (*Knight et al., 2012*). This method takes advantage of the fact that neuronal activation results in a cascade of signaling events culminating in the phosphorylation of the ribosomal protein S6 (pS6). These phosphorylated ribosomes can be immunoprecipitated from mouse brain homogenates, thereby enriching the sample for RNAs selectively expressed in the activated neuronal population (*Figure 1B*, left panel). After polysome immunoprecipitation, RNA extraction, and sequencing, RNAs enriched relative to total RNA have been shown to mark activated neurons (*Knight et al., 2012*; *Azevedo et al., 2019*; *Stern et al., 2019*). We applied this method to identify markers for neurons in the lateral septum that are activated after acute restraint stress relative to control naïve mice (*Figure 1B*). The enrichment for each gene was calculated as the number of reads in the immunoprecipitated RNA (IP) relative to the total input RNA (IP/INP; *Figure 1B*). As expected based on data from immediate-early gene mapping, we found enrichment for activity-related genes (*Fosb* and *Fos*, *Figure 1B*, middle panel) in the samples from mice exposed to acute restraint stress (see *Figure 1A*, LS and *Figure 1—figure supplement 1A*). Next, we analyzed the data to identify other genes that were significantly enriched (>2-fold or greater with a q-value <0.05) in the samples from mice exposed to acute restraint stress (RS) compared to control mice (q-value <0.05, *Figure 1B* and *Supplementary file 2*). We found enrichment for several markers, including the gene for the neuropeptide neurotensin (*Nts*, a 2.41-fold increase compared to control samples; *Schroeder et al., 2019*). We confirmed the enrichment using qPCR and found a 3.6-fold enrichment in neurotensin mRNA in the samples from the lateral septum of mice exposed to acute restraint stress compared to samples from control mice (*Figure 1C*, Unpaired Student's t-test, p<0.05).

We further confirmed the activation of *Nts*-expressing neurons by acute restraint stress by analyzing the colocalization of *Nts* and *Fos* mRNA expression in the LS using in situ hybridization (*Figure 1D*). We found significant colocalization of *Fos* and *Nts* in the LS after mice were subjected

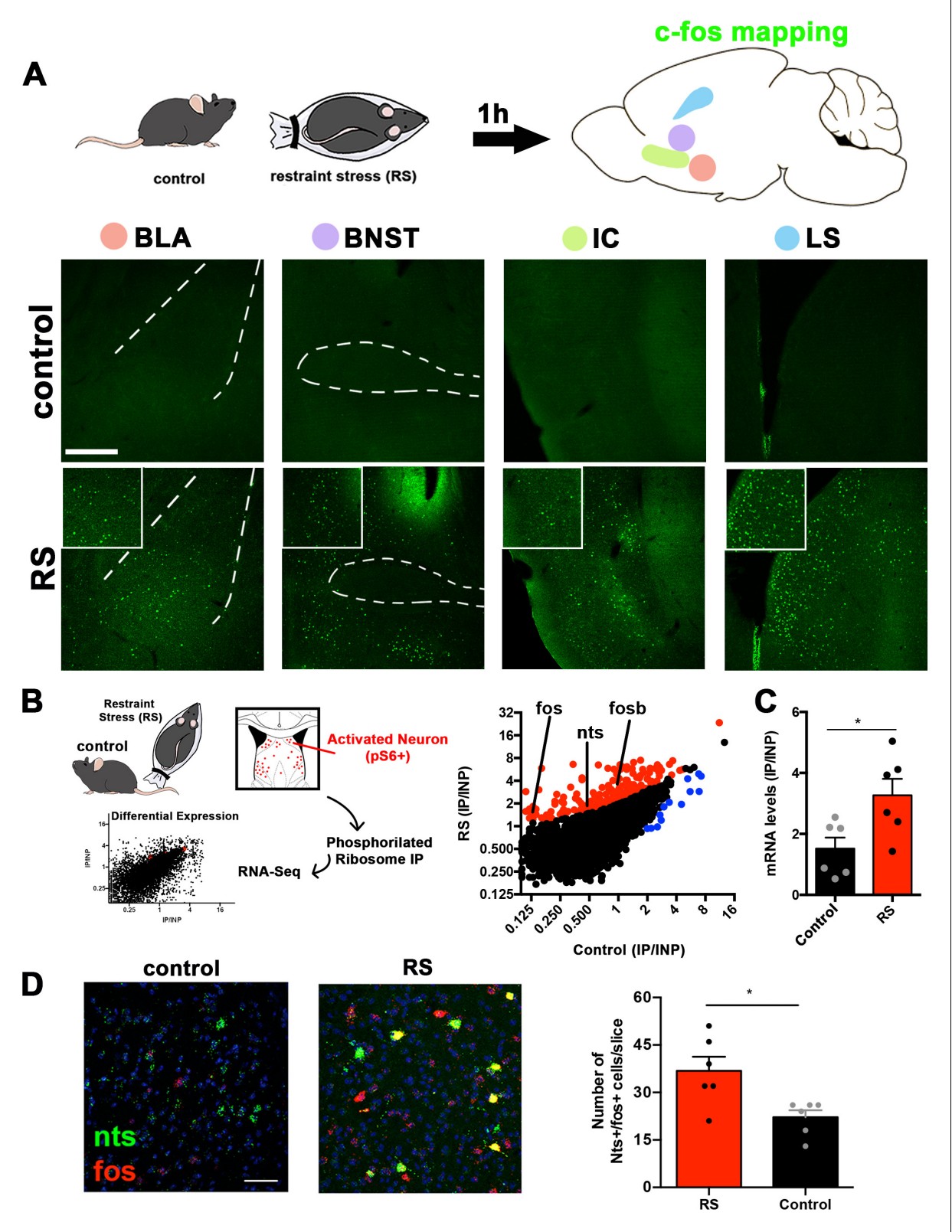

**Figure 1.** Identification of lateral septum neurons activated by acute restraint stress. (**A**) Activity mapping of the whole brain using c-fos staining in control (naïve) and restraint-stressed (RS) mice. Areas with the highest c-fos expression (green) difference are shown: the basolateral amygdala (BLA), bed nucleus of the stria terminals (BNST), insular cortex (IC), and lateral septum (LS). Insets show a 2× digital zoom of c-fos expression. Scale bars, 50 μm. (**B**) Activity-based transcriptomics (PhosphoTrap) of the lateral septum of control (naïve) and restraint-stressed mice. Middle panel, plot depicting

*Figure 1 continued on next page*

*Figure 1 continued*

the average IP/INP value (log2) of all genes analyzed in control and restraint stress (RS) samples. Enriched genes (>2-fold; red) and depleted genes (<2-fold; blue) are shown, and activation or target gene markers are shown: *Nts* (neurotensin), *Fos*, and osb (n = 2). All genes depicted have q-values <0.05 as calculated by Cufflinks. (C) mRNA levels of *Nts* evaluated using Taqman qPCR in control and restraint stress (RS) samples: control (black bars) and RS (red bars) are compared (n = 6). Unpaired Student's t-test, *p<0.05, t = 2.692, df = 10. Data are represented as mean ± SEM. (D) In situ hybridization showing the expression of *Nts* (green) and *Fos* (red) mRNA in males exposed to acute restraint stress (RS) or naïve (control). Quantification of the number of *Nts+/Fos+* cells per slice/mice are shown, n = 6, Unpaired Student's t-test, *p<0.05, t = 2.962, df = 10. Scale bar, 25 μm. Data are represented as mean ± SEM.

The online version of this article includes the following figure supplement(s) for figure 1:

**Figure supplement 1.** LS^NTS neurons are activated by acute restraint stress.

to restraint stress compared to controls (~37 cells in stressed mice vs. 22 cells in non-stressed control mice; Unpaired Student's t-test, p<0.05, *Figure 1D*). We also found that the increased colocalization between *Fos* and *Nts* in stressed mice was due to both an increase in the levels of *Fos* in *Nts* neurons and a concomitant induction of *Nts* mRNA during stress (*Figure 1—figure supplement 1B*). These results suggest that the activity of Nts neurons as well as the expression of the *Nts* gene in the LS is regulated by acute restraint stress.

## LS^Nts neurons regulate food intake and body weight

In humans and rodents, the sensory cues conveying the perception of danger during stressful events have lasting behavioral consequences. We thus assayed food intake after 1 hr of acute restraint stress and observed a highly significant decrease in consumption for the ensuing 2 hrs (*Figure 2A*, Two-way ANOVA with Bonferroni post-hoc, *p<0.05, ***p<0.001). Because we observed activation of Nts neurons following restraint stress, we next assessed the functional consequences of activating LS^Nts neurons on food intake and other behaviors. We injected an AAV encoding a Cre-dependent excitatory DREADD (Designer Receptor Exclusively Activated by Designer Drugs [*Armbruster et al., 2007*]) into the LS of Nts-Cre mice. As expected, injection of clozapine-N-oxide (CNO) into these mice increased c-fos expression in the LS (*Figure 2—figure supplement 1A*, Unpaired Student's t-test, p<0.01) and significantly decreased food intake compared to mice treated with CNO after injection of a control Cre-dependent mCherry AAV into Nts-cre mice (*Figure 2B*, red bars vs. black bars, Two-way ANOVA with Bonferroni post-hoc, ***p<0.001) as well as compared with saline-injected mice expressing the DREADD construct (*Figure 2—figure supplement 1B*). This robust decrease in food intake was evident after 1 hr and persisted for 4, 8, and 24 hr after a single CNO injection (*Figure 2—figure supplement 1C*). Chronic stimulation of *Nts* neurons for 3 d in the LS also significantly decreased food intake (*Figure 2C*, Two-way ANOVA with Bonferroni post-hoc, **p<0.01, ****p<0.0001) and significantly reduced body weight relative to the aforementioned controls (*Figure 2D*, Unpaired Student's t-test, p<0.05).

Next, we tested whether acute restraint stress could override homeostatic feeding in animals that were provided with food after an overnight fast. We found that fasted mice that had also been exposed to acute restraint stress for 1 hr consumed significantly less food during refeeding than controls (*Figure 2E*, Unpaired Student's t-test, *p<0.05), indicating that acute stress can override the hyperphagia that follows an overnight fast. Consistent with this, chemogenetic activation of LS^Nts neurons in mice also suppressed food intake in mice that were refed after that an overnight fast (*Figure 2F*, Unpaired Student's t-test, *p<0.05).

To investigate whether inactivation of LS^Nts neurons could have an opposite effect on food intake and stimulate consumption, we injected Cre-dependent AAV expressing the inhibitory hM4Di receptor or a control AAV expressing mCherry into the LS of Nts-cre mice. Acute inhibition of LS^Nts neurons led to an increase in food intake after CNO injection compared to mCherry controls in the first 2 hr (*Figure 2G*, Two-way ANOVA with Bonferroni post-hoc, ***p<0.001) and at 8 and 24 hr (*Figure 2—figure supplement 1D*). As expected, food intake was unchanged after saline-injection into mice expressing the inhibitory DREADD (*Figure 2—figure supplement 1E*). Chronic inhibition of Nts neurons in the LS for 3 d produced an increase in food intake that was significant on the 3rd day (*Figure 2H*, Two-way ANOVA with Bonferroni post-hoc, *p<0.05) but did not increase body weight (*Figure 2I*, Unpaired Student's t-test, p=0.83). These results show that LS^Nts neurons negatively regulate food intake in mice.

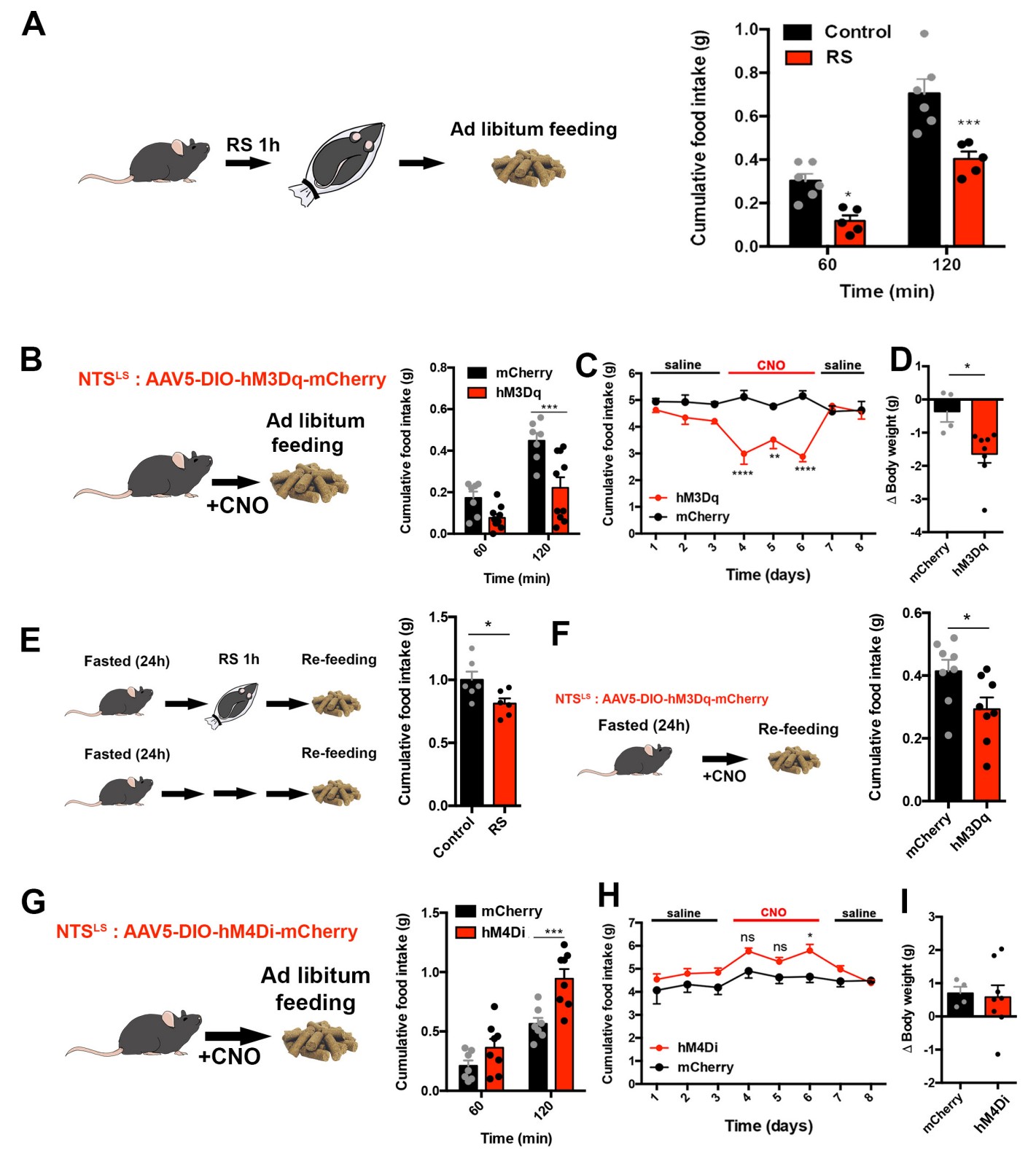

**Figure 2.** LS[Nts] neurons regulate food intake and body weight in mice. (**A**) Left panel, experimental scheme. Right panel, food intake, measured in grams (g), in control (black) and RS (red) mice after 60 and 120 min following RS. n = 5(RS), n = 6(control), Two-way ANOVA with post-hoc Bonferroni correction, Time: $F_{(1,9)}=175.9$, p<0.0001; Subject: $F_{(1,9)}=16.86$, p=0.003; Interaction: $F_{(1,9)}=4.975$, p=0.0527. (**B**) Left panel, experimental scheme. Nts-cre mice were injected with an AAV encoding the hM3Dq activatory DREADD receptor in the LS. Middle panel, cumulative food intake, measured in

*Figure 2 continued on next page*

*Figure 2 continued*

grams (g) of control mCherry (black) and hM3Dq (red) expressing Nts-cre mice after 60 and 120 min following CNO injection. n = 7(mCherry), n = 10 (hM3Dq), Two-way ANOVA with post-hoc Bonferroni correction, Time: $F_{(1,15)}=55.46$, p<0.0001; Subject: $F_{(1,15)}=12.84$, p=0.0027; Interaction: $F_{(1,15)}=5.45$, p=0.034. (C) Daily food intake, measured in grams (g) daily during injections of saline (days 1–3, 7–8) or CNO (days 4–6) in control mCherry (black) or hM3Dq (red) expressing Nts-cre mice. n = 8; Two-way ANOVA with post-hoc Bonferroni correction, Time: $F_{(7,72)}=2.871$, p=0.01; Subject: $F_{(1,72)}=46.92$, p<0.0001; Interaction: $F_{(7,72)}=6.48$, p<0.0001. (D) Body weight (delta), measured in grams (g) after 3 d of daily CNO injections in control mCherry (black) or hM3Dq (red) expressing Nts-cre mice. n = 8; Unpaired Student's t-test, *p<0.05, t = 2.903, df = 10. (E) Left panel, experimental scheme. Right panel, cumulative food intake, measured in grams (g), in control naïve mice (control, black) or mice that underwent restraint stress (RS, red), measured 2 hr after re-feeding after an overnight fast, n = 6. Unpaired Student's t-test: T(10)=2.427, p=0.0356. (F) Left panel, experimental scheme. Right panel, cumulative food intake, measured in grams (g), in control mCherry (black) or hM3Dq (red) expressing Nts-cre mice, measured 2 hr following re-feeding after an overnight fast, n = 8, Paired Student's t-test, *p<0.05, t = 3.171, df = 7. (G) Left panel, experimental scheme. Right panel, cumulative food intake, measured in grams (g), in control mCherry (black) and hM4Di (red) expressing Nts-cre mice after 60 and 120 min following saline injection. n = 7(mCherry), n = 8(hM4Di). Two-way ANOVA with post-hoc Bonferroni correction, Time: $F_{(1,19)}=415.8$, p<0.0001; Subject: $F_{(1,13)}=1.042$, p=0.326; Interaction: $F_{(4,52)}=2.839$, p=0.0333. (H) Daily food intake, measured in grams (g) after daily injections of saline (days 1–3, 7–8) or CNO (days 4–6) in control mCherry (black) or hM4Di (red) expressing Nts-cre mice. n = 7; Two-way ANOVA with post-hoc Bonferroni correction, Time: $F_{(7,96)}=3.949$, p=0.0008; Subject: $F_{(1,96)}=18.95$, p<0.0001; Interaction: $F_{(7,96)}=0.876$, p=0.5285. (I) Body weight (delta), measured in grams (g) after 3 d of daily CNO injections in control mCherry (black) or hM4Di (red) expressing Nts-cre mice. n = 8; Unpaired Student's t-test, p=0.83, t = 0.2168, df = 10. Data are represented as mean ± SEM.

The online version of this article includes the following figure supplement(s) for figure 2:

**Figure supplement 1.** LS$^{Nts}$ neurons regulate feeding.

**Figure supplement 2.** LS$^{Nts}$ neurons do not regulate anxiety-like behaviors.

Interestingly, LS$^{Nts}$ chemogenetic activation did not alter other behaviors often associated with stressful conditions. This was evaluated by assaying open field thigmotaxis, general locomotion (*Figure 2—figure supplement 2A–C*), neophobia or anxiety-related behaviors when animals were in an open-field with novelty (*Figure 2—figure supplement 2D*), in an elevated plus maze (*Figure 2—figure supplement 2E–G*), or during novelty-suppressed feeding (*Figure 2—figure supplement 2H*). Activation of LS$^{Nts}$ neurons failed to elicit an effect on any of these other behaviors. In aggregate, these data show LS$^{Nts}$ neurons are activated by restraint stress and reduce food intake, but do not play a discernible role in anxiogenic behaviors.

## Molecular profiling of LS$^{Nts}$ neurons reveal the expression of a diversity of molecules and receptors

The majority of the neurons in the LS are GABAergic and a number of neuropeptides and specific receptors are expressed in this region in addition to neurotensin (*Lein et al., 2007*). In order to identify possible signals that regulate LS$^{Nts}$ neurons, we molecularly profiled LS$^{Nts}$ neurons by injecting a Cre-dependent adeno-associated virus expressing a GFP-tagged ribosomal protein (AAV-Introvert-EGFPL10a) into the LS of Nts-Cre mice (*Nectow et al., 2017*). Polysomes were immunoprecipitated with an anti-GFP antibody to profile LS$^{Nts}$ neurons using viral-based translating ribosome affinity purification followed by RNA-seq (viral -TRAP; *Figure 3A*). The enrichment for each gene was calculated as the number of reads in the immunoprecipitated RNA (IP) relative to the total input RNA (IP/INP; *Figure 3B*). Several genes were significantly enriched in the immunoprecipitated fraction (IP) including *Gfp, Nts, Sst, Glp1r, Cartpt,* and *Mc3r* (*Figure 3B and C*, *Supplementary file 3*). We confirmed the expression of these transcripts using qPCR of the precipitated RNAs (*Figure 3C*) and in situ hybridization which also showed that ~80% of septal Nts neurons co-express Slc32a1 (*Vgat*), a marker of GABAergic neurons (*Figure 3D*, Unpaired Student's t-test, **p<0.01). We also verified the co-expression of *Nts* with *Glp1r, Sst, Cartpt,* and *Mc3r* using in situ hybridization, and confirmed that 70% of LS$^{Nts}$ neurons express Glp1r, 50% express Sst, 40% express Cartpt, and 20% express Mc3r (*Figure 3E–I*).

Many of the genes co-expressed with Nts in LS$^{Nts}$ neurons are known to regulate feeding behavior via effects on other brain areas, including Cartpt (*Kristensen et al., 1998*), Glp1r (*Kanoski et al., 2016*), Mc3r (*Marks et al., 2006*), and Sst (*Luo et al., 2018*). For this reason, we asked whether activation of LS neurons expressing these genes could replicate the effects of LS$^{Nts}$ chemogenetic activation. Similar to chemogenetic activation of LS$^{Nts}$ neurons, activation of Glp1r neurons within the LS with the Glp1r agonist exendin-4 (*Figure 3J and K*) resulted in a robust decrease in food intake after

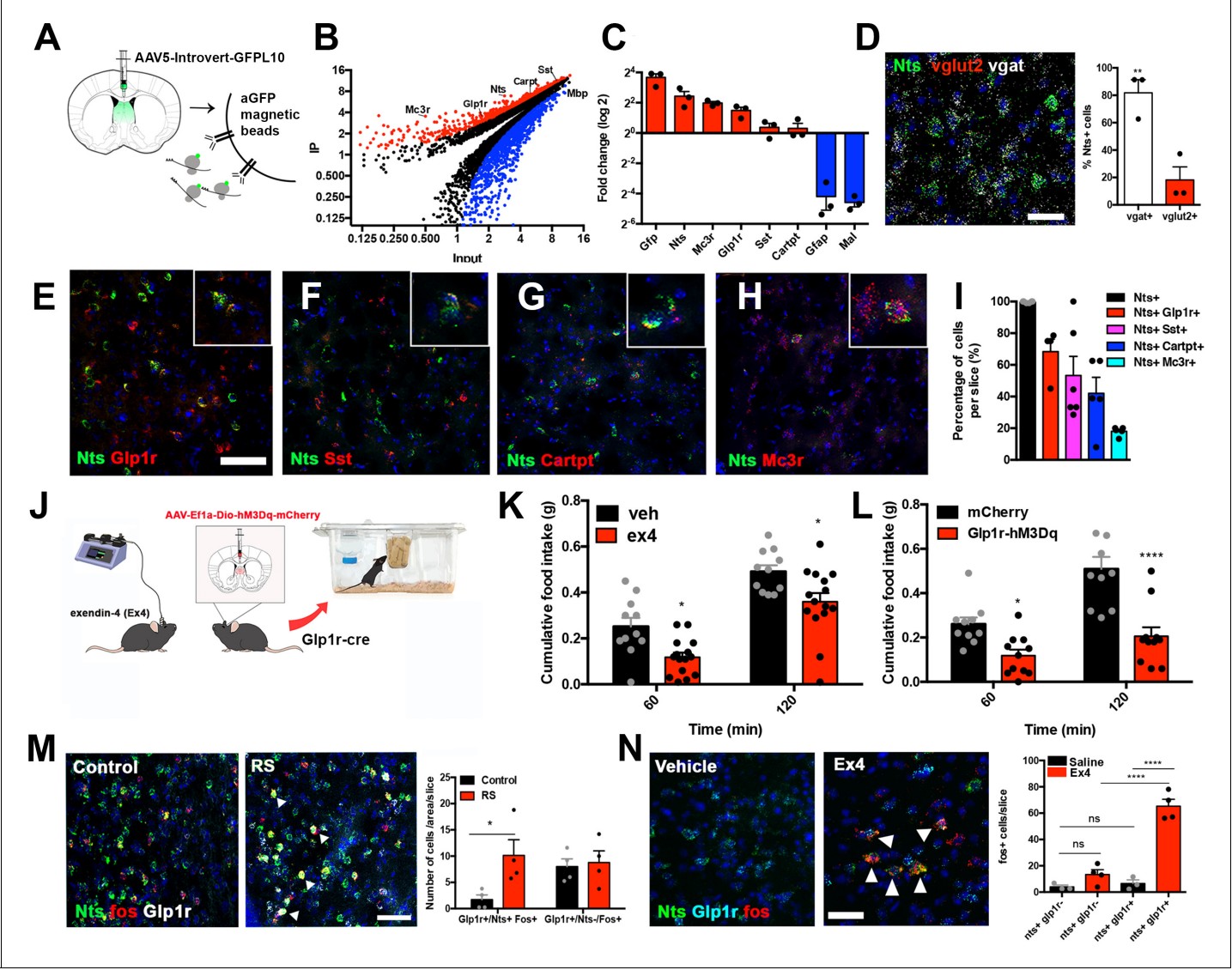

**Figure 3.** Molecular and functional profiling of LS^Nts neurons. (**A**) Experimental scheme of the viralTrap experiment. Nts-cre mice are injected with an AAV expressing cre-dependent GFPL10 (AAV-Introvert–GFPL10) into the LS and then GFP+ polysomes are immunoprecipitated. (**B**) Plot depicting the average IP value and average input values (log2) of all genes analyzed. Enriched genes (>2-fold; red) and depleted genes (<2-fold; blue) are shown (q-value <0.05), and selected markers are shown: *Mc3r*, *Glp1r*, *Nts*, *Sst*, *Cartpt*, and *Mbp* (n = 3). All genes depicted have q-values <0.05 as calculated by Cufflinks. (**C**) Average fold of change (IP/INP Log2) assessed using Taqman qPCR of enriched positive control markers Gfp and Nts, significantly enriched genes *Mc3r*, *Glp1r*, *Sst*, and *Cartpt* (red) and depleted negative control markers *Gfap* and *Mal* (blue) (n = 3 biological replicates). (**D**) Left panel, representative in situ hybridization image of *Nts* (green), *Vglut2* (red), *Vgat* (white), and DAPI (blue). Right panel, quantification of *Nts* cells which express *Vgat* (white) and *Vglut2* (red). n = 3, Unpaired Student's t-test p=0.009; t(4)=4.791. Scale bars, 50 μm. (**E**) Representative in situ hybridization image of *Nts* (green) and *Glp1r* (red). 2× digital zoom is presented in the inset. Scale bars for panels e-h, 50 μm. (**F**) Representative in situ hybridization image of *Nts* (green) and *Sst* (red). Closer magnification is inset. 2× digital zoom is presented in the inset. (**G**) Representative in situ hybridization image of *Nts* (green) and *Cartpt* (red). 2× digital zoom is presented in the inset. (**H**) Representative in situ hybridization image of *Nts* (green) and *Mc3r* (red). 2× digital zoom is presented in the inset. (**I**) Quantification of the percentage of *Nts* cells (black) expressing *Nts/Glp1r* (red), *Sst* (magenta), *Cartpt* (blue), and *Mc3r* (cyan). n = 3(*Nts*), n = 4(*Nts/Glp1r, Nts/Mc3r*), n = 6(*Nts/Sst*), n = 5(*Nts/Cartpt*) (**J**) Left panel, experimental scheme. Wild-type mice are injected with exendin-4 (Ex4) directly into the LS or Glp1r-cre mice with viral expression of hM3Dq activatory DREADD receptors are tested for cumulative food intake. (**K**) Cumulative food intake measured in grams (g) of mice following Vehicle (black) and Ex4 (red) injection in the LS. n = 11 (Vehicle), n = 15(Ex4), Two-way ANOVA with post-hoc Bonferroni correction, Time: F(1,24)=100.8, p<0.0001; Treatment: F(1,24)=11.54, p=0.0024; Interaction: F(1,24)=0.002; p=0.963. (**L**) Cumulative food intake measured in grams (g) of control mCherry (black) and hM3Dq (red) expressing Glp1r-cre mice following CNO injection. Two-way ANOVA with post-hoc Bonferroni correction. n = 10(mCherry), n = 11(hM3Dq). Time: F(1,19)=40.67, p<0.0001; Subject: F(1,19)=20.68, p=0.0002; Interaction: F(1,19)=9.407, p=0.0063. (**M**) Representative images of *Nts* (green), *Fos* (red), and *Glp1r* (white) in control and restraint stress (RS) mice. Right, quantification of co-localization between *Nts*, *Fos*, and *Glp1r*+ cells in control (black) and RS (red) mice. n = 4, Two-

*Figure 3 continued on next page*

Figure 3 continued

way ANOVA with post-hoc Bonferroni correction, *p<0.05, Row factor: $F_{(1,12)}=1.460$, p=0.2502, Column Factor: $F_{(1,12)}=5.056$, p=0.0441, Interaction: $F_{(1.12)}=3.494$, p=0.0862. Scale bars, 50 μm. (**N**) Left, representative in situ hybridization image of *Nts* (green), *Glp1r* (blue), and *Fos* (red) of mice injected with vehicle or the Glp1r agonist Exendin-4 (Ex4). White arrows point to cells co-expressing *Nts*, *Glp1r*, and *Fos*. Right, quantification of *Fos*+ cells expressing *Nts* and/or *Glp1r* in saline (black) or Ex4 (red) injected mice, n = 3(Control), n = 4 (Ex4); Two-way ANOVA with Bonferroni correction, Subject: $F_{(1,10)}=71.7$, p<0.0001; Treatment: $F_{(1,10)}=45.88$, p=0.0001; Interaction: $F_{(1,10)}=37.58$, p<0.0001. Scale bars, 25 μm. Data are represented as mean ± SEM.

The online version of this article includes the following figure supplement(s) for figure 3:

**Figure supplement 1.** Regulation of feeding and/or anxiety by Cartpt-, Glp1r- and Sst-expressing neurons in the lateral septum.

2, 12, and 24 hr (*Figure 3—figure supplement 1A–B*). In addition, chemogenetic activation of Glp1r neurons by injecting the activatory hM3Dq DREADD into the LS of Glp1r-cre mice also reduced food intake (Two-way ANOVA with Bonferroni post-hoc, *p<0.05, ****p<0.0001). Neither control mice expressing the DREADD, nor mice injected with saline or with mCherry-expressing AAVs, exhibited any change in food intake (*Figure 3K and L*; *Figure 3—figure supplement 1A–C*). By contrast, food intake was unchanged after infusion of the Mc3r agonist (gamma-melanocyte-stimulating hormone; y-msh) into the septum or activation of Cartpt or Sst neurons by injecting an activatory DREADD (hM3Dq) into Cartpt-cre and Sst-cre animals (*Figure 3—figure supplement 1D–F*). Activation of these other LS^Nts subpopulations (*Figure 3—figure supplement 1G–I*) also had no effect on arm exploration as assessed using an elevated plus maze (*Figure 3—figure supplement 1J–L*), suggesting that they do not elicit anxiety.

Glp-1 signaling has been shown to exert an anorectic effect and blocking of Glp1r signaling using specific antagonists has been shown to blunt stress-induced anorexia (*Terrill et al., 2018*). After acute restraint stress, we observed significantly increased expression of *Fos* only in LS^Nts neurons that also co-express *Glp1r*, while *Fos* expression was unchanged in *Glp1r* neurons that do not express *Nts* (*Figure 3M*, Two-way ANOVA with Bonferroni post-hoc, *p<0.01). Furthermore, infusion of Exendin-4 (ex4), a Glp1r agonist, into the LS increased *Fos* expression in ~65% of LS^Nts neurons co-expressing *Glp1r* (*Figure 3N*, Two-way ANOVA with Bonferroni post-hoc, ****p<0.0001), but not in *Nts* neurons that do not co-express *Glp1r* (*Figure 3N*). These results demonstrate the heterogeneity of the Nts population in the LS and identify a stress-responsive subpopulation of LS^Nts neurons co-expressing Glp1r that negatively regulates food intake and further shows that Glp-1 (glucagonlike peptide-1) signaling to this subpopulation of LS^Nts neurons regulates their activity.

## Mapping the downstream projections of LS^Nts neurons

Next, we set out to map the functional axonal projections of LS^Nts neurons by injecting a cre-dependent AAV expressing mCherry into the LS of Nts-cre mice. We found dense axonal projections of these neurons in the hypothalamus, in particular the lateral hypothalamus (LH, *Figure 4A*). We also mapped this projection by injecting a Cre-dependent anterograde herpes simplex virus, H129ΔTK-tdTomato into the LS of Nts-cre mice (*Figure 4B*). A similar LS to LH projection was identified when mapping axonal projections from Glp1r-cre mice expressing a Cre-dependent mCherry from an AAV injected into the LS (*Figure 4—figure supplement 1A*). The LH is a well-established brain region regulating food intake (*Morrison et al., 1958*) and we thus tested whether specific activation of an LS^Nts →LH projection would lead to a similar decrease in food consumption as activation of LS^Nts cell bodies. We injected a Cre-dependent AAV expressing a channelrhodopsin (Chr2) into the LS of Nts-cre mice and implanted a fiber above the LH (*Figure 4—figure supplement 1B*). Optogenetic activation of this pathway with blue light led to a significant increase in c-fos expression in the LS^Nts neurons (~181 c-fos+ neurons for Chr2 vs. ~8.5 for mCherry) and a subset of neurons in the LH (~128 c-fos+ neurons for Chr2 vs. ~35 for mCherry, *Figure 4—figure supplement 1C–D*, Unpaired Student's t-test, **p<0.01). We then used a 1 hr OFF-ON-OFF protocol to evaluate food intake before, during, and after light stimulation of LS^Nts terminals in the LH and observed a reversible and significant decrease of food intake in mice expressing ChR2 compared to controls expressing mCherry (*Figure 4C*, Two-way ANOVA with post-hoc Bonferroni correction, *p<0.05). Furthermore, similar to the effect of activating the entire LS^NTS population of neurons, activation of the LH-projecting LS^Nts neurons led to a ~50% decrease in food intake compared to controls (*Figure 4C*, right panel,

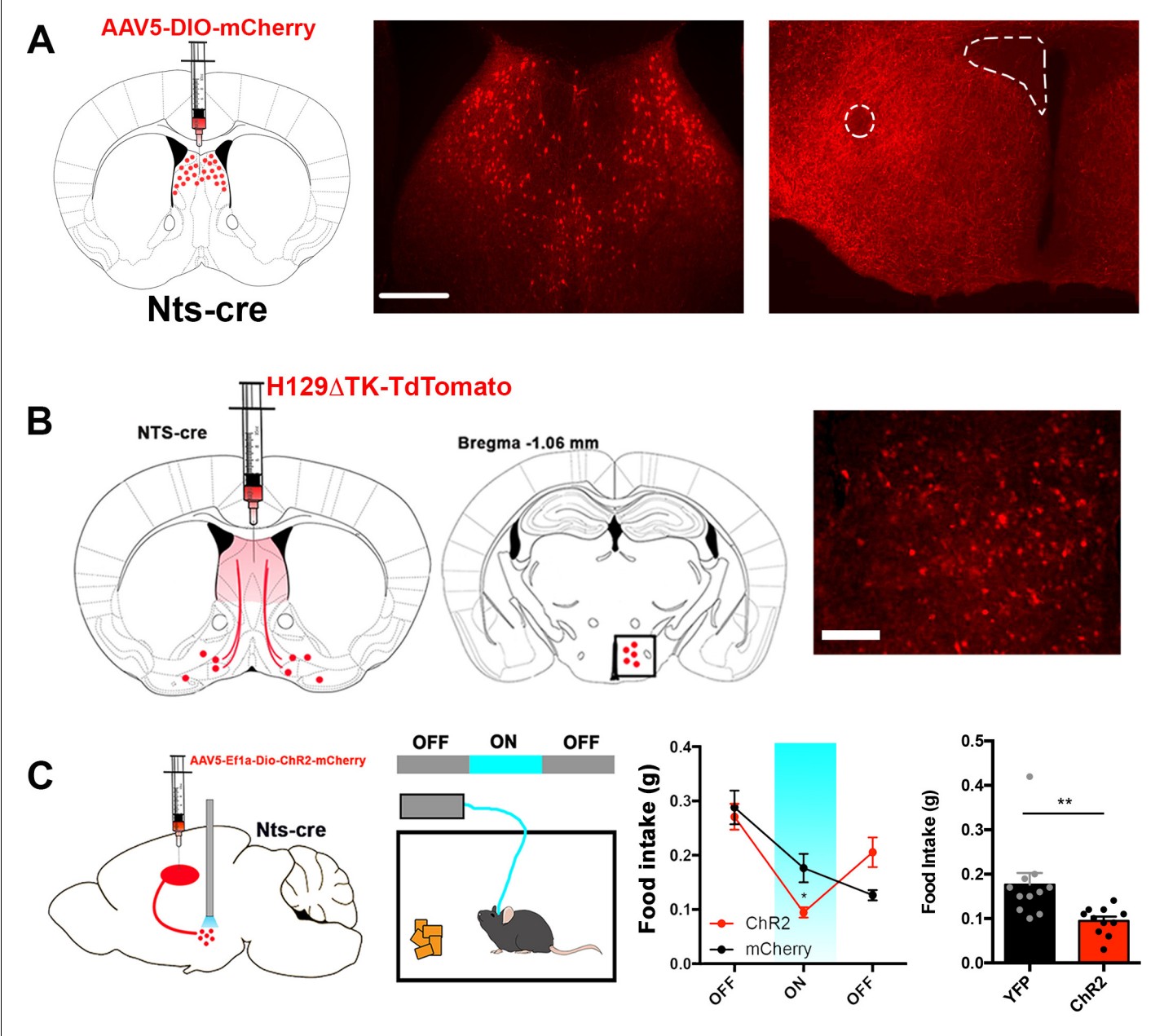

**Figure 4.** LS$^{Nts}$- > LH projection regulates food intake in mice. (**A**) Representative images of the LS injection site (left) and terminals in the lateral hypothalamus (left) of Nts-cre mice injected with a control mCherry reporter virus. Atlas images above the representative images depict the area of magnification. Scale bar, 50 μm. (**B**) Representative image of cells labeled in the lateral hypothalamus (right) of Nts-cre mice injected with the anterograde tracer H129ΔTK-tdTomato in the LS. Middle, atlas image depicting the area of magnification to the right. Scale bar, 25 μm. (**C**) Left, experimental scheme depicting mice injected with AAV expressing Chr2 or control mCherry into the LS and an optic fiber positioned over the lateral hypothalamus. Middle, food intake was measured in an OFF-ON-OFF paradigm. Right, food intake measured in grams (g) of control mCherry (black) or Chr2 (red) expressing mice before (OFF), during (ON) and after (OFF) laser stimulation. n = 11, Two-way ANOVA with post-hoc Bonferroni correction, Epoch: F(2,40)=38.47, p<0.0001; Subject: F(1,20)=0.069, p=0.796; Interaction: F(2,40)=10.94, p=0.0002. Data are represented as mean ± SEM. The online version of this article includes the following figure supplement(s) for figure 4:

**Figure supplement 1.** LS$^{Nts}$ → LH does not regulate anxiety or locomotion.

Unpaired Student's t-test, p<0.01). We also evaluated whether the activation of LH-projecting LS^Nts neurons had any effect on locomotion or anxiety and found no differences during open field or elevated plus maze tests (*Figure 4—figure supplement 1E–G*).

## LS^Nts neurons are selectively tuned to active escape

We followed up on the observation that c-fos in LS^Nts neurons is increased following acute restraint stress by asking how LS^Nts neurons respond to different types of stress . We assayed the dynamics of LS^NTS activation using in vivo calcium recordings after injecting AAVs expressing a Cre-dependent GCaMP6s into the LS of Nts-cre mice . Neural activity was monitored using in vivo fiber photometry in freely behaving animals while time-locking behavior with changes in calcium signals (*Gunaydin et al., 2014*; *Figure 5* and *Figure 5—figure supplement 1A*). GCaMP6s expression and optical fiber placement were confirmed by immunohistochemistry following each of the experiments (*Figure 5—figure supplement 1B*). We first assessed activity during acute manual immobilization by manually restraining mice for a 1 min period. Based on the c-fos data (*Figure 1*), we expected to see an increase in calcium activity while the animal was being actively restrained. Surprisingly, we did not see a change in the calcium signal when the animal was rendered completely immobile. Rather, we only found increases in the calcium signal when animals were actively struggling to escape immediately either before or after the manual restraint. The reduced activity LS^Nts neurons was seen both when the animals were completely immobilized or when they had ceased struggling (*Figure 5A*, One-way ANOVA, multiple comparisons with Bonferroni post-hoc, *p<0.05). We then recorded LS^Nts calcium signals in animals that were subjected to 1 min of being suspended by their tails during which time the animals struggle and actively try to escape. Consistent with the previous results, we observed significant activation of LS^Nts neurons while the animals were suspended and actively struggled (*Figure 5B*, Paired Student's t-test, *p<0.05).

We thus considered the possibility that LS^Nts neurons are specifically activated by stressful situations where the animal can escape, that is, flight. To test this further, we exposed mice to a simulated 'predator' using a remote-controlled robotic spider (*Figure 5C*, left panel). In this paradigm, mice are habituated to an open field arena containing a homemade opaque acrylic nest that serves as a protected 'safe zone'. The remote-controlled robotic spider was placed into the arena and after 5 min of habituation, the spider movement was manually controlled so that it only moved when mice left the acrylic nest. We observed that the forward movement of the robotic spider toward the mouse induced a rapid retreat and led the mice to immediately return to the inside the acrylic nest (*Video 1*). We simultaneously used in vivo fiber photometry to monitor the activity of LS^Nts neurons during this simulated predator attack and found that movement of the robotic spider toward the mouse led to strong activation of LS^Nts neurons that persisted until the mouse had returned to the nest (*Figure 5C*, Paired Student's t-test, *p<0.05). Similarly, we observed a rapid and sustained increase in the calcium signal when the nest was removed and mice were exposed to the robotic spider (*Figure 5D*, Paired Student's t-test, *p<0.05). In aggregate, these results show that LS^Nts neurons are activated by stressful stimuli in which the animal's response includes active movement such as struggling or flight (active coping strategies) and further suggest that these neurons are not activated by stressful stimuli associated with immobility or freezing, that is, passive coping responses.

To further confirm the tuning of LS^Nts to stress stimuli that involved active coping, we monitored the activity of LS^Nts neurons in mice during a contextual fear conditioning protocol (CFC), in which mice are trained to associate a novel environment with an unpredicted foot shock. This task enables a comparison of different types of stress responses because during the training phase, the foot shock leads to active movement away from the shock source, while during testing, exposure to the contextual cue results in freezing behavior (*Figure 5—figure supplement 1G–H*). Moreover, unlike restraint, which is imposed by the experimenter, freezing is self-initiated. Consistent with the prior results, we found that during the training session, LS^Nts calcium signals were increased following the foot shock, time-locked with movement as the mice jumped away from the shock (*Figure 5E and F*, Paired Student's t-test, p<0.05). However, during the recall test the following day, when mice exhibited freezing behavior after being placed in the chamber that they associated with a foot shock, the LS^Nts calcium signal was significantly decreased compared to the period before the onset of freezing (*Figure 5G*, Paired Student's t-test, p<0.05). Taken together, these data are consistent with the possibility that the activity of these neurons is tuned to active, but not passive, responses to stressful stimuli. They also indicate that these neurons remain active until the animal is in an environment that

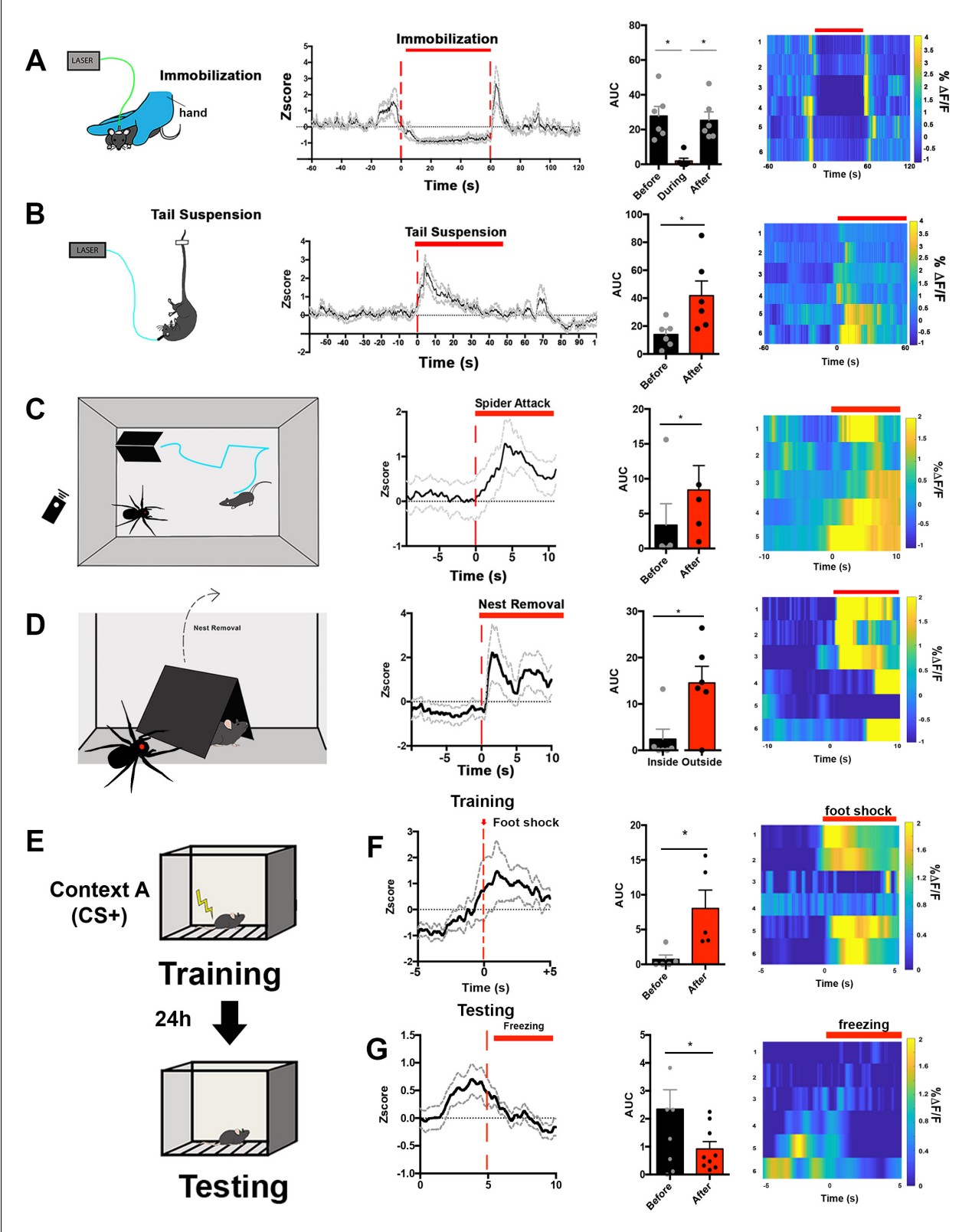

**Figure 5.** LS[Nts] neurons are specifically activated by stressful situations involving active coping. (**A**) Left panel, experimental scheme: mice were manually immobilized for the period of 1 min. Ca[+2] activity was recorded before, during, and after the period of manual immobilization. Middle, Average GCaMP6s z-score (black lines) and SEM (gray dotted lines) across all recordings and time-locked to immobilization start. Right middle panel, quantification of the area under the curve of Ca[+2] transients before, during, and after immobilization. One-way ANOVA with repeated measures and

*Figure 5 continued on next page*

*Figure 5 continued*

post-hoc Bonferroni correction, F(1.833, 9.165)=9.888, p=0.0057. Right panel, heat maps represent the average % ΔF/F from each GCaMP6s recordings time-locked to immobilization start. n = 6. (B) Left panel, experimental scheme: mice were tail suspended for the period of 1 min. $Ca^{+2}$ activity was recorded before, during, and after the period of tail suspension. Middle, average GCaMP6s z-score (black lines) and SEM (gray dotted lines) across all recordings and time-locked to tail suspension start. Right middle panel, quantification of the area under the curve of $Ca^{+2}$ transients before, during, and after tail suspension. Paired Student's t-test, t = 2.659, df = 5. Right panel, heat maps represent the average % ΔF/F from each GCaMP6s recordings time-locked to tail suspension start. n = 6. (C) Left panel, experimental scheme: mice were exposed to an open field arena containing a robotic, remote-controlled spider predator and a designated safe 'nest' (black triangular hut). $Ca^{+2}$ activity was recorded during the whole session. Middle, average GCaMP6s z-score (black lines) and SEM (gray dotted lines) across all recordings and time-locked to the robotic spider attack. Right middle panel, quantification of the area under the curve of $Ca^{+2}$ transients before and after the robotic spider attack. Paired Student's t-test, t = 3.507, df = 4. Right panel, heat maps represent the average % ΔF/F from each GCaMP6s recordings time-locked to the robotic spider attack. n = 5. (D) Left panel, experimental scheme: the designated safe 'nest' (black triangular hut) was removed and mice that had been attacked by a robotic spider predator was exposed to the open field containing the spider. $Ca^{+2}$ activity was recorded during the whole session. Middle, average GCaMP6s z-score (black lines) and SEM (gray dotted lines) across all recordings and time-locked to the removal of the nest. Right middle panel, quantification of the area under the curve of $Ca^{+2}$ transients before and after the removal of the nest. Paired Student's t-test, t = 3.305, df = 5. Right panel, heat maps represent the average % ΔF/F from each GCaMP6s recordings time-locked to the removal of the nest. n = 6. (E) Experimental scheme for contextual fear conditioning: mice were exposed to an operant chamber where they received a mild foot shock (training) and after 24 hr mice were re-exposed to the operant chamber (testing). $Ca^{+2}$ activity was recorded during training and testing. (F) First panel, average GCaMP6s z-score (black lines) and SEM (gray dotted lines) across all recordings and time-locked to the foot shock received on the training day. Second panel, quantification of the area under the curve of $Ca^{+2}$ transients before and after the foot shock. Paired Student's t-test, t = 3.230, df = 4. Third panel, heat maps represent the average % ΔF/F from each GCaMP6s recordings time-locked to the freezing behavior. n = 6. (G) First panel, average GCaMP6s z-score (black lines) and SEM (gray dotted lines) across all recordings and time-locked to freezing behavior on the testing day. Second panel, quantification of the area under the curve of $Ca^{+2}$ transients before and after the freezing behavior. Paired Student's t-test, t = 2.833, df = 8. Third panel, heat maps represent the average % ΔF/F from each GCaMP6s recordings time-locked to the freezing behavior. n = 6. Data are represented as mean ± SEM.

The online version of this article includes the following figure supplement(s) for figure 5:

**Figure supplement 1.** LS[Nts] neurons are not selectively tuned to feeding or movement.

is either perceived to be 'safe' or until the animal freezes as part of a passive coping strategy. By contrast, we did not find any significant temporal correlation between calcium signals in LS[Nts] neurons and feeding (*Figure 5—figure supplement 1C–D*) or voluntary movement in an open field arena in the absence of danger signals (*Figure 5—figure supplement 1E–F*).

## Discussion

Stress elicits a pleiotropic set of adaptive physiologic and behavioral responses enabling animals to rapidly respond to threatening situations. These responses include suppression of other behaviors such as feeding while animals initiate defensive behaviors. Accordingly, the inhibition of consummatory behaviors when animals are threatened decreases an animal's vulnerability to predation (*Petrovich et al., 2009*; *Kunwar et al., 2015*). Many brain regions are activated after stressful stimuli including the LS which shows increased c-fos expression under different stressful conditions (*Sheehan et al., 2004*). We set out to identify these stress-activated neurons using an unbiased screen to identify molecular markers of LS neurons that are activated by stress. Our study identified a novel population of neurons in the LS that express Nts (LS[Nts] neurons). We further showed that these neurons are specifically activated by active escape in response to stress and that chemogenetic activation of LS[Nts] neurons can suppress food intake and body weight via a projection to the LH. These neurons respond to stress and specifically regulate feeding, but not other components of the stress

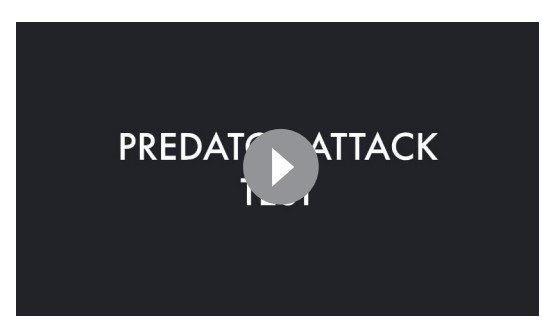

**Video 1.** Representative video of the simulated 'predator' test. The mouse is placed in the open field with a designated safe zone. A robotic spider 'predator' is introduced and 'attacks' as the mouse exits the safe zone. Calcium recordings were taken throughout the entire task.
https://elifesciences.org/articles/58894#video1

response. These data thus delineate a molecularly defined limbic circuit selectively tuned to active escape and that acts to suppress food intake in mice.

These data are consistent with a number of prior studies suggesting that the LS plays a role in mediating elements of a stress response (*Singewald et al., 2011*; *Bondi et al., 2007*) though the cell types in the LS that mediate this response had not been identified. In our study, we also characterized the calcium dynamics of the LS^Nts neurons response using in vivo calcium recordings. We first performed a manual immobilization experiment while recording from LS^Nts neurons. We found that LS^Nts neurons were not active during immobilization and were only active before or after the immobilization period when animals could struggle. This finding was surprising because our initial results from experiments placing animals in a plastic decapicone led to increased c-fos expression in the LS. There are a few possibilities that might explain these conflicting results. First, the temporal activation of c-fos is imprecise compared to measuring calcium dynamics and did not allow us to discern whether the LS^Nts neurons were activated while the animal was moving prior to or after the stress versus when they were completely immobile. Second, these two experiments were subtly different in that restraint stress using the decapicone may allow subtle movements such as struggling/squirming while the manual immobilization did not. We encountered technical difficulties while performing the same restraint experiment using the decapicones during in vivo recordings due to the volume of the fiber-optic implant that hindered our experimental approach. Thus, in our in vivo calcium recording studies, mice were manually restrained by the experimenter's hand and we observed that mice did not actively struggle during this type of restraint. This also raises the possibility that either manual restraint does not cause significant stress or this type of stress elicits effects independently of LS^Nts neurons and is thus encoded differently. To further resolve this, we employed additional stress paradigms to better understand the dynamics of Nts activation in the lateral septum. We found that LS^Nts neurons are specifically activated only in response to stressful stimuli associated with active movement (i.e. flight) and not with stressful situations associated with freezing/immobility. Thus, LS^Nts neurons were activated during active escape in a predator-attack paradigm, during tail suspension, and during the startle response to a foot shock. By contrast, the immobility associated with manual restraint, or freezing after fear conditioning did not lead to the activation of these neurons. This suggested that different neural populations within the LS may be specially tuned to different aspects of the stress response (sensory, movement, among others). Further studies will be needed to determine whether other populations are activated by complete immobility or immobility (i.e. freezing) and define whether or not these populations intersect with the pathways activated by LS^Nts neurons.

We also found that LS^Nts neurons, in particular, do not regulate anxiety-like behaviors but instead reduce food intake suggesting that they link specific stressful stimuli to reduced food intake. Thus, LS^Nts neuronal activation appears to be tuned to active coping mechanisms that dispense with voluntary food intake until appropriate countermeasures to mitigate the stressful event have been enacted. For example, this circuit would be useful for suppressing the urge to forage when predators are in the vicinity, thus serving an important evolutionary function. Consistent with this, activation of LS^Nts neurons suppresses food intake even in animals that have been deprived of food, although future studies are needed to confirm the sufficiency of these neurons for stress-induced reductions in feeding by inhibiting the LS^Nts neurons during the stressful stimulus. Many of the stress paradigms used during in vivo calcium recordings were performed acutely which we found to be insufficient to elicit a significant impact on food intake. However, several of the stressors that were used, such as foot shock, can reduce food intake if applied chronically (*Halataei et al., 2011*; *Rostamkhani et al., 2016*). If LS^Nts sufficiency is confirmed, neural plasticity and/or genetic alterations within this circuit (e.g. by mutations in the Nts gene as observed in *Lutter et al., 2017* or changes in 'allostatic load'; *McEwen and Akil, 2020*) could contribute to persistent maladaptive eating strategies, such as those observed in eating disorders or in other cases obesity. Presumably, other neural circuits mediate the effects of stressful stimuli that lead to freezing, raising the possibility that different circuits have evolved to process different types of stress, and that different types of chronic stress might be associated with different disorders.

The LS neuronal population is mainly composed of GABAergic neurons although there are also some glutamatergic neurons (Sunkin, 2013). In this study, we also explored the molecular diversity of LS^Nts neurons. Our findings suggest that the LS^Nts neuronal population is heterogeneous and co-express a variety of molecules, including Glp1r. Consistent with our results, a previous report showed that the infusion of a Glp1r antagonist (Ex9) directly into the LS diminished the anorexigenic effects of

stress, suggesting that Glp-1 signaling in LS contributes to this response (*Terrill et al., 2018*). The LS is innervated by pre-proglucagon (PPG) neurons in the brainstem expressing Glp-1 (*Trapp and Cork, 2015*) and these neurons have also been shown to mediate stress-induced hypophagia (*Holt et al., 2019*). In aggregate, these data suggest that LS[Nts] neurons that co-express Glp1r provide a link between brainstem circuits conveying stress to key output pathways to regulate feeding. The LS also receives inputs from many cortical and limbic areas, including the hippocampus (*Tingley and Buzsáki, 2018*; *Azevedo et al., 2019*; *Jakab and Leranth, 1990*) and in turn sends projections to many other brains areas (*Deng et al., 2019*) including the hypothalamus, hippocampus, entorhinal cortex, thalamus, and amygdala. Further studies to map which brain areas convey inputs to LS[Nts] neurons and control their excitability are currently underway.

In summary, these data show that LS[Nts] neurons play an important integratory function linking stress to food suppression and serving as an important integrator between the limbic system and the hypothalamus. The identification of these neurons thus provides an entry point for understanding how complex inputs —in this case food deprivation versus danger—compete to generate an appropriate and adaptive response. In addition, these findings may in time illuminate why in some cases stress can elicit a pathologic response such as in the case of eating disorders. These and other data further suggest that higher-order, limbic brain regions integrate diverse sensory information to modulate subcortical feeding circuits in health and possibly disease (*Azevedo et al., 2019*; *Newmyer et al., 2019*; *Land et al., 2014*).

# Materials and methods

## Key resources table

| Reagent type (species) or resource | Designation | Source or reference | Identifiers | Additional information |
|---|---|---|---|---|
| Genetic reagent (*Mus musculus*) | Mouse: *Nts*-cre | Jackson Laboratories | Stock#017525 | N/A |
| Genetic reagent (*Mus musculus*) | Mouse: *Cartpt*-cre | Jackson Laboratories | Stock#028533 | N/A |
| Genetic reagent (*Mus musculus*) | Mouse: *Sst*-cre | Jackson Laboratories | Stock#013044 | N/A |
| Genetic reagent (*Mus musculus*) | Mouse: *Glp1r*-cre | Jackson Laboratories | Stock#029283 | N/A |
| Antibody | Anti-phospho ribosomal protein 6 pSer244/pSer247 (Rabbit polyclonal) | Invitrogen | Cat#44–923G, RRID:AB_2533798 | (1:1000) |
| Antibody | Anti-GFP (Chicken polyclonal) | Abcam | Cat#ab13970, RRID:AB_300798 | (1:1000) |
| Antibody | 19C8 and 19F7, GFP monoclonal antibodies | Memorial Sloan-Kettering Monoclonal Antibody Facility | Custom order | (50 ug) |
| Antibody | Anti-mCherry (Chicken polyclonal) | Abcam | Cat# ab205402 | (1:1000) |
| Antibody | Mouse anti-cfos (Rabbit monoclonal) | Cell Signaling | Cat#2250S, RRID:AB_2247211 | (1:500) |
| Antibody | Anti-rabbit IgG Alexa 488 (Goat polyclonal) | Invitrogen | Cat#A11008, RRID:AB_143165 | (1:1000) |
| Antibody | Anti-rabbit IgG Alexa 594 (Goat polyclonal) | Invitrogen | Cat#A11072, RRID:AB_142057 | (1:1000) |
| Antibody | Anti-chicken IgG Alexa 488 (Goat polyclonal) | Invitrogen | Cat#A11039, RRID:AB_142924 | (1:1000) |
| Antibody | Anti-chicken IgG Alexa 594 (Goat polyclonal) | Invitrogen | Cat#A11042, RRID:AB_142083 | (1:1000) |
| Recombinant DNA reagent | AAV5-EF1a-DIO-YFP | UNC Vector Core | N/A | N/A |

*Continued on next page*

*Continued*

| Reagent type (species) or resource | Designation | Source or reference | Identifiers | Additional information |
|---|---|---|---|---|
| recombinant DNA reagent | AAV5-EF1a-DIO-hChR2(H134R)-YFP | UNC Vector Core | N/A | N/A |
| Recombinant DNA reagent | AAV5-EF1a-DIO-hM3D (Gq)-mCherry | UNC Vector Core | N/A | N/A |
| Recombinant DNA reagent | AAV5-EF1a-DIO-mCherry | UNC Vector Core | N/A | N/A |
| Recombinant DNA reagent | AAV5-EF1a- DIO-hM4D (Gi)-mCherry | UNC Vector Core | N/A | N/A |
| Recombinant DNA reagent | AAV5-DIO-GCaMP6s | Addgene | Cat#510882 | N/A |
| Recombinant DNA reagent | HSV-lsl-tdTomato | *Lo and Anderson, 2011* | N/A | N/A |
| Recombinant DNA reagent | AAV5-Introvert-GFPL10a | *Nectow et al., 2017* | N/A | N/A |
| Sequence-based reagent | PrimeTime Standard qPCR Assay for NTS | idtDNA | Mm.PT.58.10351472 | N/A |
| Sequence-based reagent | PrimeTime Standard qPCR Assay for ActB | idtDNA | Mm.PT.47.5885043.g | N/A |
| Sequence-based reagent | Probe for NTS (C1) | ACDBio | Cat# 420448 | N/A |
| Sequence-based reagent | Probe for cFos (C2) | ACDBio | Cat#506921-C2 | N/A |
| Sequence-based reagent | Probe for Glp1r (C3) | ACDBio | Cat#418851-C3 | N/A |
| Sequence-based reagent | Probe for Sst (C3) | ACDBio | Cat#404631-C3 | N/A |
| Sequence-based reagent | Probe for Mc3r (C3) | ACDBio | Cat#412541-C3 | N/A |
| Sequence-based reagent | Probe for Cartpt (C3) | ACDBio | Cat#432001-C3 | N/A |
| Sequence-based reagent | Probe for vGAT (C2) | ACDBio | Cat#319191-C2 | N/A |
| Sequence-based reagent | Probe for vGLUT2 (C3) | ACDBio | Cat#319171-C3 | N/A |
| Peptide, recombinant protein | NBL10 recombinant peptide | Chromotek | Cat# GT-250 | 100 ng/mL |
| Commercial assay or kit | RNAeasy Mini kit | Qiagen | Cat#74104 | N/A |
| Commercial assay or kit | RNAscope Fluorescent Multiplex Reagent Kit V2 | ACDbio | Cat#323100 | N/A |
| Chemical compound, drug | Clozapine-N-oxide(CNO) | Tocris Biosciences | Cat#4936; CAS: 34233-69-7 | 1 mg/kg |
| Software, algorithm | Tophat | Basespace | https://basespace.illumina.com/apps/ | N/A |
| Software, algorithm | Cufflinks | Basespace | https://basespace.illumina.com/apps/ | N/A |
| Software, algorithm | SYNAPSE | Tucker-Davis Technologies | https://tdt.com | N/A |
| Software, algorithm | MATLAB | Mathworks | https://mathworks.com | N/A |
| Software, algorithm | ImageJ | NIH | https://imagej.nih.gov/ij/ | N/A |
| Software, algorithm | GraphPad Prism 5.0 | GraphPad | https://www.graphpad.com/scientific-software/prism/ | N/A |
| Software, algorithm | Ethovision 9.0 | Noldus | https://noldus.com | N/A |

## Animals and breeding

C57BL/6J (Stock# 000664), *Nts*-Cre (*Leinninger et al., 2011*; Stock# 017525), *Sst*-Cre (Stock# 028864), *Cartpt*-Cre (Stock# 028533), and *Glp1r*-cre (Stock# 029283) mice were obtained from Jackson Laboratories and housed according to the guidelines of the Rockefeller's University Animal Facility under protocol 17105 and protocol 18039. Male and female mice between 8 and 20 weeks of age were used throughout this study and all animal experiments were approved by the Rockefeller University IACUC, according to NIH guidelines. Male and female mice were used for *Nts*-cre, *Sst*-cre, *Cartpt*-cre, and *Glp1r*-cre experiments. In experiments that used Cre+ and Cre– mice, littermates were used. No differences were determined based on sex and the number of males and females was counterbalanced during experiments. Mice were kept on a 12 hr/12 hr light/dark cycle (lights on at 7:00 a.m.) and had access to food and water ad libitum, except when noted otherwise. Mice were housed together (4–5 mice/cage) for most experiments, except feeding experiments (one mouse/cage). All feeding experiments used standard rodent chow pellets.

## Stereotaxic injections

All AAVs used in this study were purchased from UNC Vector Core or Addgene, except where noted. AAV5-DIO-hM3Dq-mCherry or AAV5-DIO-Chr2-mCherry was used for activation studies. AAV5-DIO-hM4Di-mCherry was used for inhibition studies. Control AAV5-DIO-mCherry virus was used for comparison and terminal projection mapping. AAV5-DIO-GCaMP6s was used for fiber photometry studies. AAV5-Introvert-GFPL10a was a gift from Alexander Nectow. H129ΔTK-TdTomato virus was a gift from David Anderson and was used as described previously (*Lo and Anderson, 2011*) Mice were anesthetized with isoflurane, placed in a stereotaxic frame (Kopf Instruments, Tujunga, CA) and bilaterally injected in the lateral septum using the following coordinates relative to bregma: AP: +0.58; ML: 0.0, DV: −3.00 (*Paxinos and Franklin, 2019*). A total of 400 nL of virus at high titer concentrations (at least $10^{11}$) were injected per site at a rate of 100 nL/min. For pharmacological experiments, cannulae were implanted in the LS at the same coordinates as above. For fiber photometry experiments, optic fiber implants (Doric, Quebec, Candada) were inserted at the same coordinates as above. For optogenetics experiments, optic fiber implants (Thor Labs, Newton, NJ) were implanted bilaterally over the LH at the following coordinates relative to bregma: AP: −1.80 mm; ML: + 1.50 mm, DV: −5.25 mm at a 10° angle (*Paxinos and Franklin, 2019*). Implants were inserted slowly and secured to the mouse skull using two layers of Metabond (Parkell Inc, Edgewood, NY). Mice were singly housed and monitored in the first weeks following optic fiber implantation. For all surgeries, mice were monitored for 72 hr to ensure full recovery. After 2 weeks, mice were used for experiments. Following behavioral procedures, mice were euthanized to confirm viral expression and fiber placement using immunohistochemistry.

## Restraint stress

Except where noted for fiber photometry studies, restraint stress consisted of 1 hr of immobilization in a decapicone (Braintree Scientific, Braintree, MA) in a separate room adjacent to the housing room. Mice were returned to their home cage following restraint for further food intake measurements or until euthanasia. For in vivo fiber photometry, restraint stress was given manually by the investigator during recordings.

## Fiber photometry

Mice were acclimated to patch cables for 5 min for 3 d before experiments. Analysis of the signal was done using the fiber photometry system and processor (RZ5P) from TDT (Tucker-Davis Technologies, Alachua, FL), which includes the Synapse software (TDT). The bulk fluorescent signals from each channel were normalized to compare across animals and experimental sessions. The 405 channel was used as the control channel. GCaMP6s signals that are recorded at this wavelength are not calcium-dependent; thus, changes in signal can be attributed to autofluorescence, bleaching, and fiber bending. Accordingly, any fluctuations that occurred in the 405 control channels were removed from the 465 channel before analysis. Change in fluorescence (ΔF) was calculated as (465 nm signal − fitted 405 nm signal), adjusted so that a ΔF/F was calculated by dividing each point in ΔF by the 405-nm curve at that time. Z-scores were calculated as $\frac{signal\ -\ signal\ median}{median\ absolute\ deviation}$ to account for

variability across animals. Behavioral variables were timestamped in the signaling traces via the real-time processor as TTL signals from Noldus (Ethovision) software (Wageningen, the Netherlands). This allowed for precise determinations of the temporal profile of signals in relation to specific behaviors. Behaviors were tested as follows. First, we tested the response to food. Mice were fasted overnight and then placed into a clean home cage with food pellets introduced 5 min later. Food approach and consumption were scored. We then tested restraint by manually restraining mice for 5 min. Tail suspension was conducted by suspending mice by their tails for 5 min. To examine the role of predator escape, mice were placed into an open field that contained a designated safe zone consisting of a homemade acrylic nest. After 5 min, a remote-controlled spider (Lattburg) was introduced into the open field and began to pursue the mouse at random intervals. After 10–15 min, the nest was removed. We scored times when the mouse fled from the spider as well as the removal of the nest. Locomotion was scored during the first 5 min in the open field arena. All behaviors were manually scored by a second investigator who did not conduct the photometry experiment.

## Chemogenetic and pharmacological feeding experiments

For acute feeding, mice were singly housed and injected with either saline or clozapine-N-oxide (CNO; Sigma) at 1 mg/kg intraperitoneally (i.p.), exendin-4 (Tocris, 1 mg/mL), or y-msh (Tocris, 150 nM) directly into the lateral septum before the dark phase and the cumulative food intake and body weight was measured at selected time periods. For chronic feeding, mice were single or double injected (every 12 hr) with CNO or saline before the dark phase and food intake and body weight were measured. For experiments that require fasting, mice were singly housed and food was removed before the dark phase (16–23 hr of fasting in total). The cumulative food intake at each time point was calculated for analysis.

## Elevated plus maze

Mice were placed at the center of a cross-shaped, elevated maze in which two arms are closed with dark walls and two arms are open and allowed to explore for 10 min. Mice were injected with CNO (1 mg/kg) 1 hr before testing. Afterward, the mice were returned to their home cage and the maze floor was cleaned in between subjects. All subjects were monitored using a camera and behavior (time spent in open and closed arms, distance, and velocity) were analyzed using Ethovison 9.0 (Noldus).

## Open field/open field with novelty

Mice freely explored an open field arena ($28 \times 28$ cm$^2$), divided into center and border regions, for 5 min. The time spent (in seconds) in the center area, were taken as measures of anxiety. To increase the anxiolytic nature of the task, a novel object was placed into the center of the arena (open field with novelty). Mice were injected with CNO (1 mg/kg) 1 hr before testing. Afterward, the mice were returned to their home cage and the maze floor was cleaned in between subjects. All subjects were monitored using a camera and behaviors (time spent in center and border, distance and velocity) were analyzed using Ethovison 9.0 (Noldus).

## Optogenetics feeding experiment

Mice were previously habituated to patch cables for 3 d before experiments. Implanted optic fibers were attached to a patch cable using ceramic sleeves (Thorlabs) and connected to 473 nm laser (OEM Lasers/OptoEngine, Midvale, UT). Laser output was verified at the start of each experiment. A blue light was generated by a 473 nm laser diode (OEM Lasers/OptoEngine) at 5–10 mW of power. Light pulse (10 ms) and frequency (10 Hz) was controlled using a waveform generator (Keysight) to activate Nts+ terminals in lateral hypothalamus. Animals were sacrificed to confirm viral expression and fiber placement using immunohistochemistry. Each feeding session lasted for 80 min and was divided into one trial of 20 min to allow the animal to acclimate to the cage and three trials of 20 min each (1 hr feeding session). During each feeding session, light was turned off during the first 20 min, on for 20 min and off again for the remaining 20 min. Consumed food was recorded manually before and after each session. To facilitate measurement, three whole pellets were added to cups and food crumbs were not recorded. Feeding bouts were recorded using a camera and the Ethovision 9.0 software (Noldus).

## PhosphoTrap (activity-based transcriptomics)

PhosphoTrap profiling experiments were performed according to *Knight et al., 2012*. Briefly, mice were separated into groups of 4–6 mice per group, restraint stressed for 1 hr and sacrificed following the stress. Naive mice were kept in their home cage until euthanasia. After euthanasia, brains were removed, and septal area were dissected on ice and pooled (4–6 brain samples per experiment replicate). Tissue was homogenized and clarified by centrifugation. Ribosomes were immunoprecipitated using 4 µg of polyclonal antibodies against pS6 (Invitrogen, #44–923G) previously conjugated to Protein A-coated magnetic beads (Thermofisher). A small amount of tissue RNA was saved before the immunoprecipitation (Input) and both input and immunoprecipitated RNA (IP) were then purified using RNAeasy Mini kit (QIAGEN, Germantown, MD) and RNA quality was checked using a RNA PicoChip on a bioanalyzer. RIN values >7 were used. Experiments were performed in triplicates for each group. cDNA was amplified using SMARTer Ultralow Input RNA for Illumina Sequencing Kit and sequenced on an Illumina HiSeq2500 platform. All neurons identified using PhosphoTRAP were validated using qPCR, in situ hybridization, and functional testing.

## Viral-trap (V-Trap)

Affinity purification of EGFP-tagged polysomes was done 3 or 4 weeks after virus injections (*Nectow et al., 2017*). Briefly, mice were separated into three biological replicate groups of 4–6 mice per group and euthanized. Brains were removed, and the lateral septum was dissected on ice and pooled. Tissue was homogenized in buffer containing 10 mM HEPES-KOH (pH 7.4), 150 mM KCl, 5 mM MgCl$_2$, 0.5 mM DTT, 100 µg/mL cycloheximide, RNasin (Promega, Madison, WI) and SUPERase-In (Life Technologies, Waltham, MA) RNase inhibitors, and complete-EDTA-free protease inhibitors (Roche) and then cleared by two-step centrifugation to isolate polysome-containing cytoplasmic supernatant. Polysomes were immunoprecipitated using monoclonal anti-EGFP antibodies (clones 19C8 and 19F7; see *Heiman et al., 2008*) bound to biotinylated-Protein L (Pierce; Thermo Fisher Scientific)-coated streptavidin-conjugated magnetic beads (Life Technologies). A small amount of tissue RNA was saved before the immunoprecipitation (Input) and both input and immunoprecipitated RNA (IP) were then purified using RNAeasy Mini kit (QIAGEN). RNA quality was checked using an RNA PicoChip on a bioanalyzer. RIN values > 7 were used. Experiments were performed in triplicates for each group. cDNA was amplified using SMARTer Ultralow Input RNA for Illumina Sequencing Kit and sequenced on an Illumina HiSeq2500 platform.

## RNA sequencing and qPCR analysis

RNA sequencing raw data was uploaded and analyzed using BaseSpace apps (TopHat and Cufflinks; Illumina) using alignment to annotated mRNAs in the mouse genome (UCSC, *Mus musculus* assembly mm10). The average immunoprecipitated (IP) and input values of each enriched and depleted genes with a q-value lower than 0.05 were plotted using GraphPad Prism (GraphPad, San Diego, CA). To validate the RNA Sequencing data, qPCR using predesigned Taqman probes (idtDNA, Coralville, IA) were used and cDNA was prepared using the QuantiTect Reverse Transcription Kit (Life Technologies). The abundance of these genes in IP and Input RNA was quantified using Taqman Gene Expression Master Mix (Applied Biosystems, Bedford, MA). Transcript abundance was normalized to beta-actin. Fold of Change were calculated using the ΔΔCt method.

## Immunohistochemistry, quantifications, and imaging

Mice were perfused, and brains were postfixed for 24 hr in 10% formalin. Brain slices were taken using a vibratome (Leica, Buffalo Grove, IL), blocked for 1 hr with 0.3% Triton X-100, 3% bovine serum albumin (BSA), and 2% normal goat serum (NGS) and incubated in primary antibodies for 24 hr at 4°C. Then, free-floating slices were washed three times for 10 min in 0.1% Triton X-100 in PBS (PBS-T), incubated for 1 hr at room temperature with secondary antibodies, washed in PBS-T and mounted in Vectamount with DAPI (Southern Biotech, Birmingham, AL). Antibodies used here were: anti-c-fos (1:500; Cell Signaling, Danvers, MA), anti-mCherry (1:1000; Abcam, Cambridge, MA) goat-anti-rabbit (Alexa 488 or Alexa 594, 1:1000; Molecular Probes), goat anti-chicken Alexa488 or Alexa594 (1:1000; Molecular Probes). Images were taken using Axiovert 200 microscope (Zeiss, White Plains, NY) or LSM780 confocal (Zeiss) and images were processed using ImageJ software

(NIH, *Schneider et al., 2012*). C-fos counts were conducted for three or more sections/animal and averaged for each animal for statistical analysis.

## Fluorescent in situ hybridization

For the examination of gene expression and Fos experiments, tissue samples underwent single molecule fluorescent in situ hybridization (smFISH). Isoflurane anesthetized mice were decapitated, brains harvested and flash frozen in aluminum foil on dry ice. Brains were stored at −80°C. Prior to sectioning, brains were equilibrated to −16°C in a cryostat for 30 min. Brains were cryostat sectioned coronally at 20 μm and thaw-mounted onto Superfrost Plus slides (25 × 75 mm$^2$, Fisherbrand). Slides were air-dried for 60–90 min prior to storage at −80°C. smFISH for all genes examined—*Fos*, *Nts*, *Sst*, *Glp1r*, *Cartpt*, *Mc3r*—was performed using RNAscope Fluorescent Multiplex Kit (Advanced Cell Diagnostics, Newark, CA) according to the manufacturer's instructions. Slides were counterstained for the nuclear marker DAPI using ProLong Diamond Antifade mounting medium with DAPI (Thermo Fisher). Sections were imaged using an LSM780 confocal (Zeiss) and processed using ImageJ software. Counts were conducted for two or more sections/animal and averaged for each animal for statistical analysis.

## Statistical analysis

All results are presented as mean ± SEM and were analyzed with Graphpad Prism software or Matlab. No statistical methods were used to predetermine sample sizes, but our sample sizes are similar to those reported in previous studies. Normality tests and F tests for equality of variance were performed before choosing the statistical test. Unless otherwise indicated, for the feeding experiments, we used a Two-way ANOVA with Bonferroni correction to analyze statistical differences. $p < 0.05$ was considered significant (*$p < 0.05$, **$p < 0.01$, ***$p < 0.001$, ****$p < 0.0001$). Image quantifications were analyzed using Unpaired Student's t-test. Fiber photometry data were analyzed using Paired Student's t-test. Animals in the same litter were randomly assigned to different treatment groups and blinded to investigators in the various experiments. Injection sites and viral expression were confirmed for all animals. Mice showing incorrect injection sites or optic fiber placement were excluded from the data analysis.

## Acknowledgements

We thank Ravi Tolwani and the staff of the Comparative Biosciences Center, Connie Zhao and the staff of the Genomics Resource Center, and Alison North and the staff of the Bioimaging Resource Center at Rockefeller University for technical assistance. We thank Alexander Nectow for the AAV-Introvert-GFPL10a virus and David Anderson for providing the H129ΔTK-tdTomato virus. We thank Christin Kosse for assistance in fiber photometry analysis. This work was funded by a Long Term HSFP Fellowship (EPA), F32DK107077 (SAS), K99DA048749 (SAS), NARSAD Young Investigator Awards from the Brain and Behavior Research Foundation (EPA and SAS), the JPB Foundation, and the Klarman Foundation (JMF).

## Additional information

### Funding

| Funder | Grant reference number | Author |
| --- | --- | --- |
| JPB Foundation | | Jeffrey M Friedman |
| Klarman Family Foundation | | Jeffrey M Friedman |
| Brain and Behavior Research Foundation | | Estefania P Azevedo Sarah A Stern |
| Human Frontier Science Program | | Estefania P Azevedo |
| National Institute of Diabetes and Digestive and Kidney Diseases | F32DK107077 | Sarah A Stern |

| | | |
|---|---|---|
| National Institute on Drug Abuse | K99DA048749 | Sarah A Stern |

The funders had no role in study design, data collection and interpretation, or the decision to submit the work for publication.

## Author contributions

Estefania P Azevedo, Conceptualization, Funding acquisition, Investigation, Writing - original draft, Project administration, Writing - review and editing; Bowen Tan, Violet Ivan, Marc Schneeberger, Katherine R Doerig, Investigation; Lisa E Pomeranz, Resources, Investigation; Robert Fetcho, Conor Liston, Software, Methodology; Jeffrey M Friedman, Supervision, Funding acquisition, Project administration, Writing - review and editing; Sarah A Stern, Conceptualization, Formal analysis, Supervision, Funding acquisition, Investigation, Writing - original draft, Project administration, Writing - review and editing

## Author ORCIDs

Estefania P Azevedo (iD) https://orcid.org/0000-0002-3149-7270
Bowen Tan (iD) http://orcid.org/0000-0003-3036-2881
Jeffrey M Friedman (iD) https://orcid.org/0000-0003-2152-0868
Sarah A Stern (iD) https://orcid.org/0000-0002-3033-8251

## Ethics

Animal experimentation: This study was performed in strict accordance with the recommendations in the Guide for the Care and Use of Laboratory Animals of the National Institutes of Health. All of the animals were handled according to approved institutional animal care and use committee (IACUC) protocols of the Rockefeller University under protocols 17105 and 18039.

## Decision letter and Author response

Decision letter https://doi.org/10.7554/eLife.58894.sa1
Author response https://doi.org/10.7554/eLife.58894.sa2

# Additional files

## Supplementary files

• Supplementary file 1. c-Fos expression in the mouse brain after restraint stress. Expression is noted as − (no expression), + (low expression), ++ (moderate expression) and +++ (high expression).

• Supplementary file 2. Phospho-TRAP differential gene expression data. Data is sorted by fold-change (log2) in restraint stress (RS) samples compared to Naïve (N) samples along with the corresponding q-values.

• Supplementary file 3. Viral-TRAP differential gene expression data. Significantly enriched (TRUE) genes are shown and are sorted by fold-change (log2).

• Transparent reporting form

## Data availability

We have deposited sequencing data in a NCBI repository (GSE154749 and GSE154758) and the supplementary files show a summary of enriched and depleted genes.

The following datasets were generated:

| Author(s) | Year | Dataset title | Dataset URL | Database and Identifier |
|---|---|---|---|---|
| Azevedo EP, Stern SA, Friedman JM | 2020 | Actvity-dependent profiling of lateral septum neurons following restraint stress | https://www.ncbi.nlm.nih.gov/geo/query/acc.cgi?acc=GSE154749 | NCBI Gene Expression Omnibus, GSE154749 |

| Azevedo EP, Stern SA, Friedman JM | 2020 | Molecular profiling of lateral septum neurotensin neurons using viralTRAP | https://www.ncbi.nlm.nih.gov/geo/query/acc.cgi?acc=GSE154758 | NCBI Gene Expression Omnibus, GSE154758 |

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
