## [Decision Letter]

Thank you for submitting your article "A limbic circuit selectively linking active escape to food suppression" for consideration by *eLife*. Your article has been reviewed by three peer reviewers, one of whom is a member of our Board of Reviewing Editors, and the evaluation has been overseen by Kate Wassum as the Senior Editor.

The reviewers have discussed the reviews with one another and the Reviewing Editor has drafted this decision to help you prepare a revised submission.

Summary:

Azevedo and colleagues use multiple techniques to identify a circuit that acts to modulate escape activity and food intake. Specifically, aided by PhosphoTrap-seq they identify a group of neurotensin (NTS)-positive neurons in the lateral septum that are activated in response to acute restraint stress. Activation of these neurons (cFos staining) is dependent on sex and estrus cycle. Using chemogenetics the authors show activation of these neurons suppresses food intake. Additionally, in vivo bulk calcium imaging data suggests these neurons are associated with escape behaviors, not necessarily the stressor itself. Furthermore, using TRAP-seq they identified the molecular identity of these NTS neurons, and showed both pharmacologically and pharmacogenetically that septal activation of the coexpressed Glp1 receptor neurons, leads to similar food suppression seen with NTS neuronal activation. Finally, using channelrhodopsin optogenetic activation of hypothalamic neurons, the authors implicate NTS neuronal projections from the lateral septum to the lateral hypothalamus in regulating food suppression. The implication of this population in stress-induced hypophagia is not novel, as a series of published papers described this phenomenon using a different molecular marker (Glp1r), which is also expressed in LS^NTS^ neurons. These studies extend beyond what has already been described in two important ways. First, they demonstrate that the anorexigenic activity of LS^NTS^ neurons is mediated via projections to the lateral hypothalamus. Second, analyses of calcium imaging data across different types of stress paradigms support the idea that LS^NTS^ neurons are only activated under stressful conditions involving active coping responses (i.e. escape). These findings will be of broad interest to researchers studying circuits regulating both feeding and stress-induced behaviors.

Essential revisions:

Sex differences, defined by restraint stress-induced c-fos and Nts expression in the LS are inconsistent with published studies in rats from 3 independent groups that restraint stress induces more c-fos in males than females (Aloisi, Zimmerman and Herdegen, 1997; Figueiredo, Dolgas and Herman, 2002; Sood, Chaudhari and Vaidya, 2018). Moreover, electrophysiological studies identified as proestrus with the highest LS neuronal firing rates; diestrus and estrus had similar levels of activity (Contreras Physiology and Behavior 2000). The failure to observe sex differences in feeding behavior raises concerns about the physiological relevance of sex differences in restraint-induced c-fos/Nts expression.

The authors frame the rationale and discussion of these experiments in the context of eating disorders. LS^NTS^ neurons are only active during the coping phase of the stress response (Figure 4) and food intake rebounds when these neurons are switched off (Figure 6C). The relevance of circuits regulating acute escape behaviors for the chronic suppression of energy intake in anorexia is not obvious and should be discussed.

The manuscript would be strengthened by removing the sex differences experiments in Figure 2 and the discussion of anorexia, and reframing the discussion around the most novel finding – the striking differences in LS^NTS^ neuronal firing in response to stressors that involving active responses vs. immobility. The presentation of the experiments should build up to the interesting and novel fiber photometry data, which should be presented at the end.

The authors claim that inconsistencies between LS^NTS^ neuronal activation in response to restraint in Figure 1 and inhibition in Figure 4 stem from differences in the ability of the mouse to escape. Calcium imaging using the restraint paradigm from Figure 1 would help to resolve this issue. A discussion of this point would be helpful.

Inhibitory DREADD experiments during stress-induced escape behavior (complementary to excitatory experiments in Figure 3D) would strengthen the study by allowing the authors to parse the requirement for these neurons in suppressing food intake vs. coping responses per se. This could be noted in the Discussion as well.

---

## [Author Response]

Essential revisions:Sex differences, defined by restraint stress-induced c-fos and Nts expression in the LS are inconsistent with published studies in rats from 3 independent groups that restraint stress induces more c-fos in males than females (Aloisi, Zimmerman and Herdegen, 1997; Figueiredo, Dolgas and Herman, 2002; Sood, Chaudhari and Vaidya, 2018). Moreover, electrophysiological studies identified as proestrus with the highest LS neuronal firing rates; diestrus and estrus had similar levels of activity (Contreras Physiology and Behavior 2000). The failure to observe sex differences in feeding behavior raises concerns about the physiological relevance of sex differences in restraint-induced c-fos/Nts expression.The authors frame the rationale and discussion of these experiments in the context of Eating Disorders. LS^NTS^ neurons are only active during the coping phase of the stress response (Figure 4) and food intake rebounds when these neurons are switched off (Figure 6C). The relevance of circuits regulating acute escape behaviors for the chronic suppression of energy intake in anorexia is not obvious and should be discussed.The manuscript would be strengthened by removing the sex differences experiments in Figure 2 and the discussion of anorexia, and reframing the discussion around the most novel finding – the striking differences in LS^NTS^ neuronal firing in response to stressors that involving active responses vs. immobility. The presentation of the experiments should build up to the interesting and novel fiber photometry data, which should be presented at the end.

We have removed the experiments showing differences between male and female animals and reframed the discussion focusing on the LS^NTS^ response to stressors. We have also moved the fiber photometry data to the end of the manuscript, as suggested by the reviewers.

The authors claim that inconsistencies between LS^NTS^ neuronal activation in response to restraint in Figure 1 and inhibition in Figure 4 stem from differences in the ability of the mouse to escape. Calcium imaging using the restraint paradigm from Figure 1 would help to resolve this issue. A discussion of this point would be helpful.

The reviewer makes a valid point but unfortunately, for technical reasons stated below, we are not able to perform this experiment. The restraint paradigm from Figure 1 is performed by placing a mouse in a decapicone (a plastic, cone shaped bag) which is then tightly to restrain the animal, with air flow through a small hole. To perform fiber photometry, we had to attach a cable to the mouse’s head. Because the implant plus the cable increases the volume which no longer fits into the decapicone (see https://www.braintreesci.com/prodinfo.asp?number=DC). For this reason, we instead used the manual restraint paradigm. We have added a statement in the second paragraph of the Discussion section explaining this technical limitation.

Inhibitory DREADD experiments during stress-induced escape behavior (complementary to excitatory experiments in Figure 3D) would strengthen the study by allowing the authors to parse the requirement for these neurons in suppressing food intake vs. coping responses per se. This could be noted in the Discussion as well.

We agree with the reviewer that this experiment would be interesting to evaluate the sufficiency of this population. Unfortunately, due to COVID-19 and current university restrictions, this experiment would require a long time to be performed. Thus, we have added a discussion about the sufficiency of this population to this behavior in the third paragraph of the Discussion section, noting that our data do not exclude the possibility that other cell populations might also contribute to the effects described herein.